# A Fusion of Geothermal and InSAR Data with Machine Learning for Enhanced Deformation Forecasting at the Geysers

Joe Yazbeck [1] and John B. Rundle [1,2,3,*]

1    Department of Physics and Astronomy, University of California, Davis, Davis, CA 95616, USA;
     jyazbeck@ucdavis.edu
2    Department of Earth and Planetary Sciences, University of California, Davis, Davis, CA 95616, USA
3    Santa Fe Institute, Santa Fe, NM 87501, USA
*    Correspondence: jbrundle@ucdavis.edu

**Simple Summary:** Earthquakes are a common occurrence at The Geysers geothermal field in California which, over the years, have led to general land sinking in the area. In our study, we explore the correlation of geothermal data and seismicity and find that the injection rate exhibits a high correlation with the number of earthquakes. Additionally, we rely on satellite imagery which measures the extent of the land subsidence and develop a machine learning model that predicts future land subsidence, finding that it has a relatively low error rate. Moreover, we incorporate geothermal data into this model and find that it performs even better. All in all, this encourages the use of machine learning models in hazard mitigation in order to minimize the potential impacts that land subsidence might bring.

**Abstract:** The Geysers geothermal field in California is experiencing land subsidence due to the seismic and geothermal activities taking place. This poses a risk not only to the underlying infrastructure but also to the groundwater level which would reduce the water availability for the local community. Because of this, it is crucial to monitor and assess the surface deformation occurring and adjust geothermal operations accordingly. In this study, we examine the correlation between the geothermal injection and production rates as well as the seismic activity in the area, and we show the high correlation between the injection rate and the number of earthquakes. This motivates the use of this data in a machine learning model that would predict future deformation maps. First, we build a model that uses interferometric synthetic aperture radar (InSAR) images that have been processed and turned into a deformation time series using LiCSBAS, an open-source InSAR time series package, and evaluate the performance against a linear baseline model. The model includes both convolutional neural network (CNN) layers as well as long short-term memory (LSTM) layers and is able to improve upon the baseline model based on a mean squared error metric. Then, after getting preprocessed, we incorporate the geothermal data by adding them as additional inputs to the model. This new model was able to outperform both the baseline and the previous version of the model that uses only InSAR data, motivating the use of machine learning models as well as geothermal data in assessing and predicting future deformation at The Geysers as part of hazard mitigation models which would then be used as fundamental tools for informed decision making when it comes to adjusting geothermal operations.

**Keywords:** hazard mitigation; machine learning; geothermal reservoirs; neural networks; InSAR

## 1. Introduction

Induced seismicity typically refers to the seismic activity arising from anthropogenic activities [1]. These activities induce various stress changes within the subsurface, directly leading to an earthquake's occurrence [2]. Fluid injection, fluid extraction, mining, and hydraulic fracturing are a few of the many activities that can lead to induced seismicity [3–5].

In many cases, this additionally leads to land subsidence in the area where these anthropogenic activities are taking place [6–9]. For example, Hejmanowski et al. [10] found that over half of the total subsidence resulting from mining-induced seismicity in the copper mining area of the Lower Silesia region of Poland happens within a few days after an earthquake's occurrence. Additionally, Deng et al. [11] found significant subsidence in the vicinity of the city of Pecos in western Texas, which they attributed to the extraction of oil and gas as well as groundwater.

In fact, land subsidence is one of the most damaging effects that accompanies these anthropogenic activities. Not only does it damage buildings and the underlying infrastructure, but it also causes deep fractures in the surface layer and alters the surface drainage pattern [12]. Moreover, land subsidence reduces the capacity of the aquifers to hold water and increases the risk of flooding [13]. Given the fact that the compaction is irreversible and permanent [14], it is crucial to monitor this phenomenon and be able to accurately measure it [15].

The Geysers geothermal field is one of the most seismically active regions in northern California [16], and it is heavily monitored by a large permanent seismic network that comprises many stations in order to record the seismic events occurring in the area [17]. Tens of micro-earthquakes typically occur each day, with the overwhelming majority of the associated hypocenters coinciding with the steam production field [18]. In fact, it is believed that the compounding and cumulative effects of these small earthquakes, which are attributed to the commercial extraction of steam and injection of condensate into the geothermal reservoir [19,20], can lead to serious infrastructural damage potentially similar to that caused by large earthquakes [21].

No major fault line directly exists at The Geysers geothermal field [22], which means that a major earthquake ($M \geq 7.0$) is unlikely to occur in the immediate area. However, the Maacama Fault, which is located about 10 km west of the field, is certainly capable of producing such an earthquake based on the measured slip rates and the actual length of the fault itself [23,24]. Hence, fluid injection at The Geysers poses a serious concern in the potential triggering of such an earthquake [25].

Another major concern is the continuous subsidence occurring at The Geysers geothermal field [26]. As stated previously, land subsidence can have a detrimental effect on the geothermal operations as well as the surrounding environment. Efforts to monitor this deformation began as early as 1972, when the United States Geological Survey (USGS) and the National Geodetic Survey set up a network of precise vertical and horizontal controls in order to monitor the effects of the geothermal production [27]. In the following 5 years, a series of first-order leveling surveys found that the field was indeed subsiding, with the greatest subsidence centered on the area with the most steam extraction at that time [27]. Further monitoring was performed by Mossop et al. [28], who used GPS receivers and found that the subsidence continued from 1977 to 1996 at a steady rate of about 5 cm/year, with peak rates of about 90 cm/year.

Damage as a result of subsidence related to geothermal operations specifically is not uncommon. To start with, geothermal operations at Cerro Prieto, which is the oldest and largest Mexican geothermal field, have caused considerable damage in the form of ground fissuring as well as severe damage to infrastructures such as irrigation canals and roads [29]. Furthermore, subsidence at the Wairakei–Tauhara geothermal system has caused casing damage to some of the eastern borefield wells as a result of the ground strains [30]. Within the Wairakei subsidence bowl, the damage also included pipelines, wells, and drains as well as roads, electric lines, and the local hotel [31]. Finally, in an attempt to stimulate an enhanced geothermal system (EGS), water was injected at high pressures into a deep well in Basel [32]. The injection, which covered a period of 6 days, resulted in more than 10,500 recorded seismic events [33]. This culminated in a magnitude 3.2 earthquake which caused an estimated CHF 7 million worth of damage to the urban area of Basel [34].

EGS efforts have taken place at The Geysers geothermal field. Traditionally, the concept of EGS is to extract heat from hot rocks that have not fractured naturally, which would

increase the number of areas where it can be applied by a great deal compared with naturally formed geothermal reservoirs [35]. After a suitable site is found, wells would be drilled into the hot rock, which becomes stimulated to produce stable fractures through which water can be injected and cycled [36]. The permeable pathways would allow the water to absorb heat, which would then be collected at the production wells [36]. However, at The Geysers, EGS projects are used to extract recoverable geothermal energy that has not been used [37]. Specifically, the Northwest Geysers EGS demonstration project aimed to stimulate a deep high-temperature reservoir by injecting water in order to increase permeability, increase reservoir pressure, mitigate corrosion, and reduce non-condensable gas concentrations [38]. Two previously abandoned wells in the northwest region of the field, Prati 32 (P-32) and Prati State 31 (PS-31), were reopened and prepared for stimulation in 2011 as an injection and production well, respectively [39]. A third well, Prati 25 (P-25), was also reopened in order to monitor the steam production [40]. The stimulation phase was able to open up new pathways for fluid flow, as evidenced by the progression of induced seismicity hypocenters [40]. Despite encountering corrosion issues in 2013, the project was largely successful in its goals [40].

Given the large areal nature of the deformation at the geothermal field, we rely on remote sensing techniques, since traditional geodetic techniques that use point-based measurements do not fully capture the entire picture [41]. Specifically, our study uses interferometric synthetic aperture radar (InSAR) to monitor and predict deformation at The Geysers. InSAR is a technique that takes two synthetic aperture radar (SAR) images over the same region taken at different times and combines them to form a map showing the displacement in the satellite's line of sight (LOS) [42]. InSAR functionality has been discussed rigorously in many studies and articles [43–45]. This geodetic technique has proven to be an essential tool in monitoring subsidence all over the world due to Sentinel-1's global coverage and abundant data supply [46,47]. For example, subsidence at the geothermal fields of the Taupo volcanic zone in New Zealand was measured using InSAR, and the measurements were found to compare well to leveling data at several fields across different time periods [48]. Furthermore, InSAR was used to study surface deformation at the Los Humeros Geothermal Field in Mexico, and the results were able to reveal a fault mechanism in the reservoir which was further corroborated by relating surface movements to volume changes within the reservoir [49].

Machine learning (ML) has become increasingly prevalent in several sectors that boast a large amount of data [50–52]. This is due to the efficiency of ML algorithms at handling such amounts of data as well as their ability to make accurate predictions and classifications [53]. Recently, ML algorithms have started being implemented using InSAR data, given the fact that there is now a large dataset of images that is publicly available as well as the fact that InSAR benefits from both spatial and temporal components [54–57]. To start with, Roberts et al. [58] developed a convolutional neural network (CNN) model that would characterize the stress field at geothermal reservoirs from InSAR surface displacements and applied it successfully at the Coso geothermal field in California, showing promise for generalization to all geothermal fields. Moreover, a K-means clustering algorithm was applied on InSAR data in order to differentiate areas of uplift and subsidence at the Brady geothermal field, and the results were validated with previous studies [59].

At The Geysers geothermal field, machine learning has been primarily used to understand changes in faulting processes using seismic signals [60]. Machine learning has also been used to optimize the parameters in ground motion models applied to The Geysers geothermal field that predict the ground acceleration and velocity [61]. However, to the authors' best knowledge, applying machine learning models to InSAR deformation time series with the aim of predicting future ground deformation has not been performed yet.

In this paper, we aim to predict land subsidence deformation that is occurring at The Geysers geothermal field due to fluid extraction. First, we perform a correlation study to look at the relationship between the seismicity in the region and the geothermal injection and production amounts. This motivates the incorporation of these geothermal data into a

machine learning model that uses InSAR to predict future deformation. We build a couple machine learning models that use long short-term memory (LSTM) and convolutional neural network (CNN) layers to make use of InSAR's spatial and temporal components. Finally, the model results are compared to a linear baseline model based on a mean squared error metric.

## 2. Materials and Methods

### 2.1. Area of Study

The Geysers geothermal field is the largest geothermal field in the world, and it is located in the Coast ranges of Northern California about 100 km north of San Francisco [28,62]. The yellow star in Figure 1 pinpoints its location on the map. The field is in a mountainous area with elevations ranging from 195 m at the bottom of Big Sulphur Creek to 1440 m at the top of Cobb Mountain, where it occasionally snows [63]. It is bounded by the Collayomi Fault to the east and the Mercuryville Fault to the west, with additional faults such as the Maacama and Healdsburg fault zones extending to the west of The Geysers [64].

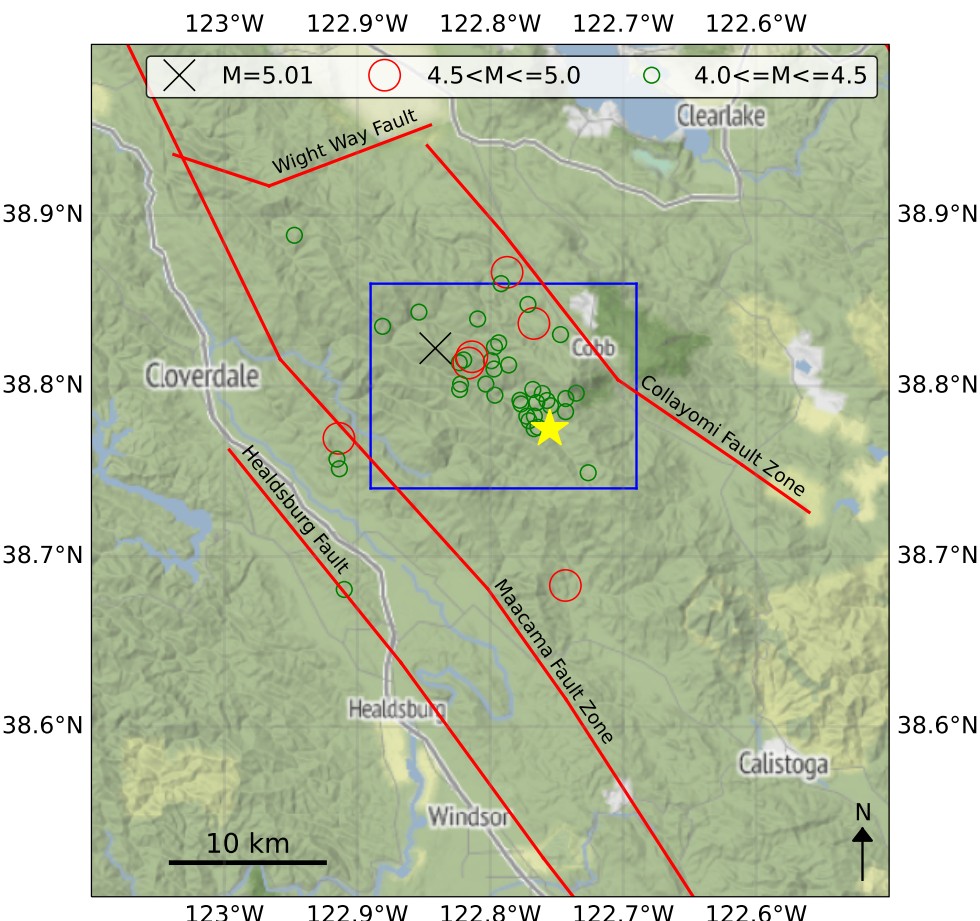

**Figure 1.** Map of The Geysers showing seismic activity in the region spanning from 1 January 1975 to 1 January 2023. The yellow star pinpoints the location of The Geysers geothermal field. The black cross indicates the location of the strongest earthquake recorded during that time period ($M_W = 5.01$ on 14 December 2016). The blue box shows the area of study for the correlation and InSAR analysis. The red lines indicate the major faults and fault zones nearby.

The existing geothermal system forms a part of the bigger Clear Lake Volcanic System that is located about 40 km north of The Geysers [65], which has been intermittently active for the last 2 million years [66]. Geothermal energy extraction began over 60 years ago when the first geothermal power plant was built in 1960 [67]. However, over time,

steam production and reservoir pressure started decreasing due to the increased use of the geothermal field [68]. Since fluid injection is known to increase the pore pressure of the rock matrix [69], water collected from rainfall, local creeks, and aquifers started being injected back into the wells [70]. Additionally, a 66 km pipeline was built in 2003 that transports treated effluents from the nearby city of Santa Rosa to supplement the water injection [71]. This resulted in an increased production rate, and the geothermal field currently produces around 800 MW of electricity, which powers about 800,000 homes in the nearby counties [60]. Given the proven effectiveness of using remote sensing techniques to track and monitor ground subsidence [72–75] as well as the ineffectiveness of point-based ground measurements at capturing the large areal nature of the problem [76], we utilize InSAR data in our analysis of the subsidence at The Geysers geothermal field.

### 2.2. Correlation Study

Our objective was to investigate the relationship between the injection and production of water and steam, respectively, and the local seismic activity at The Geysers. The seismicity data were acquired from the United States Geological Survey (USGS) catalog, while the geothermal injection and production data were acquired from the Department of Conservation of California. We started by plotting the monthly seismicity in addition to the monthly amounts of injected water and steam produced as shown in Figure 2. Visually, it appears that the injection amount exhibits some relation to the seismicity for the magnitude threshold used. In general, it seems that as one increases, the other increases, and vice versa. Additionally, we plotted the annual version of the same plot in Figure 3, where a similar relation can be seen for the injection and seismicity.

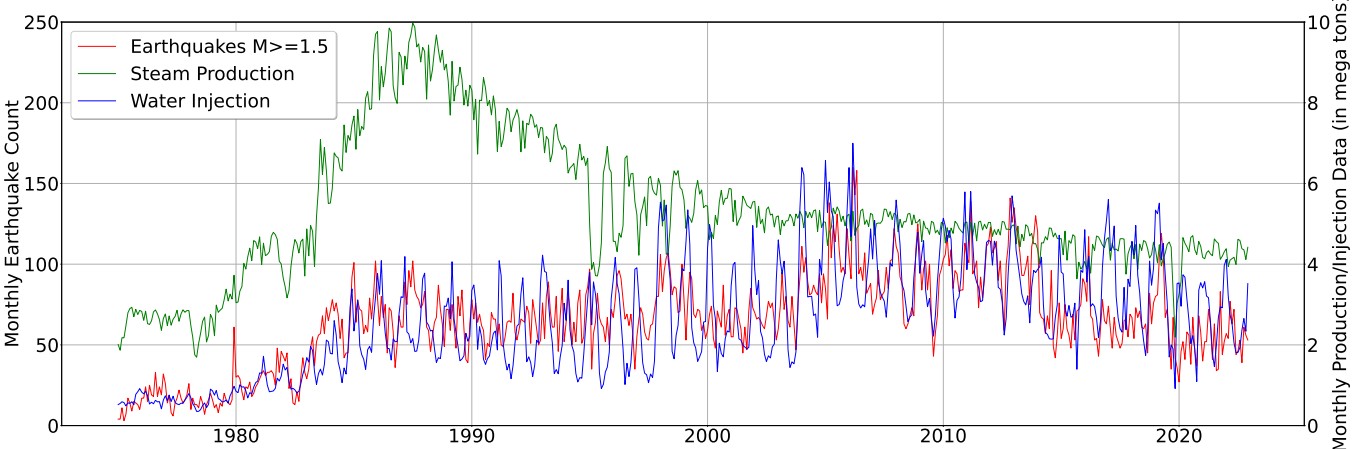

**Figure 2.** A history of the monthly seismicity count (M ≥ 1.5) at The Geysers along with the injection and production amounts from 1975 to 2023.

In order to quantify the relation between these two variables, we relied on the Pearson correlation coefficient and the Spearman correlation coefficient, which would indicate to us how correlated the injection and seismicity are and to what extent. The Pearson correlation coefficient is given by [77]

$$r_{xy} = \frac{\sum_{i=1}^{n}(x_i - \bar{x})(y_i - \bar{y})}{\sqrt{\sum_{i=1}^{n}(x_i - \bar{x})^2}\sqrt{\sum_{i=1}^{n}(y_i - \bar{y})^2}} \tag{1}$$

where $x$ and $y$ are the injection amount and seismicity count, respectively, with $\bar{x}$ and $\bar{y}$ being the associated means, $n$ is the number of data points, and $r_{xy}$ is the resultant Pearson correlation coefficient whose value ranges from −1 to 1. This measures the degree of the linear relationship between the two variables, with positive values indicating a positive linear relationship and vice versa. The magnitude of the correlation coefficient indicates

the strength of the linear relationship. However, one must note that the Pearson correlation coefficient is sensitive to outliers as it uses the exact values of the given data points [78].

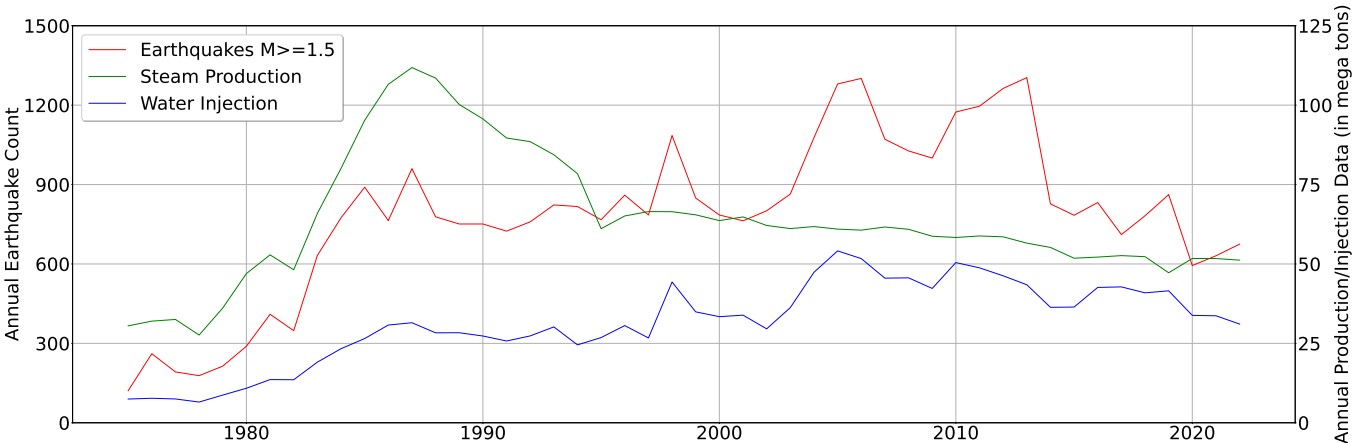

**Figure 3.** A history of the annual seismicity count (M $\geq$ 1.5) at The Geysers along with the injection and production amounts from 1975 to 2023.

The Spearman correlation coefficient is given by [79]

$$\rho = 1 - \frac{6\sum_{i=1}^{n} d_i^2}{n(n^2 - 1)} \tag{2}$$

where $d_i$ is the the difference between the ranks of the corresponding values in the two variables and $\rho$ is the resultant Spearman correlation coefficient that also varies from $-1$ to 1. In this case, the coefficient measures the monotonic relationship, with positive values indicating a positive relationship and vice versa. The magnitude similarly measures the strength of that relationship. Since the coefficient relies on the ranks of the variables rather than the values themselves, the Spearman correlation coefficient is much less sensitive to outliers than the Pearson correlation coefficient [80]. Hence, using both coefficients in conjunction with each other will give us a clearer picture of the underlying relationship between the injection and seismicity at The Geysers.

*2.3. Data Preprocessing and Baseline Model*

We created an InSAR time series using an open-source Python package called LiCS-BAS [81–83], which utilizes InSAR images that have been processed through an automatic InSAR processor called LiCSAR. These InSAR images are publicly accessible through the Centre for the Observation and Modelling of Earthquakes, Volcanoes and Tectonics: Looking inside the Continents from Space (COMET-LiCS) web portal [84,85]. We used a frame ID of 115D_05066_252015 to run LiCSBAS and generated an InSAR time series from mid-2017 to early 2022 using the default parameters. Atmospheric data were also supplemented to LiCSBAS using the Generic Atmospheric Correction Online Service for InSAR (GACOS) [86–89]. The perpendicular baseline $b_\perp$ for most interferograms was between $-50$ m and $+50$ m, with the greatest value being $+150$ m. The small baseline inversion algorithm was applied to produce displacement time series in the line of sight (LOS) of the satellite after filtering the InSAR images and building the suitable network. A velocity map was built by taking the least squares fit for each pixel's displacement time series as shown in Figure 4, where one can see that The Geysers lies in a subsiding region, indicated by the negative velocities.

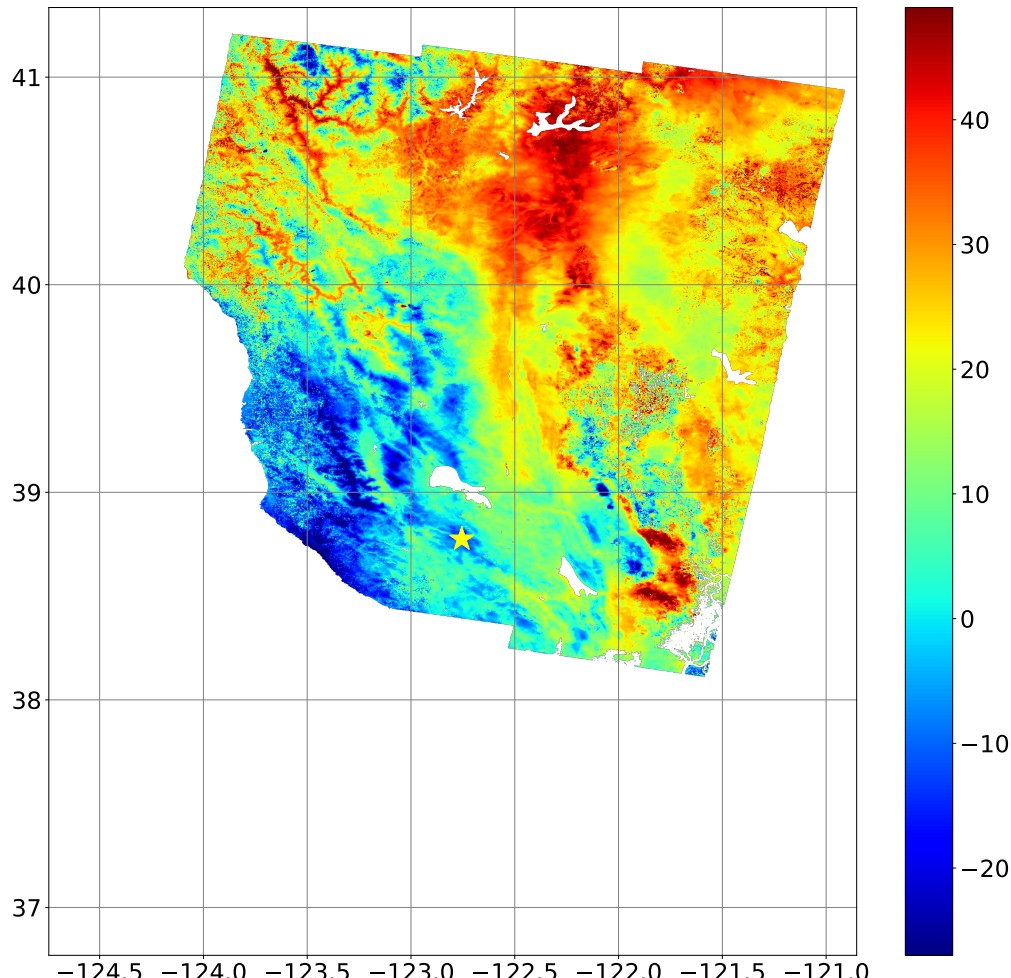

**Figure 4.** The resulting velocity map formed using LiCSBAS. Color indicates the speed in the line of sight of the satellite in units of mm/year. The yellow star indicates the location of The Geysers geothermal field.

In order to facilitate processing and training, we downsized our InSAR dataset by clipping it to the study region shown in Figure 1. This resulted in a series of 88 images with $121 \times 201$ pixels centered on The Geysers geothermal field covering roughly 90 m$^2$. Then, we performed a series of preprocessing steps to clean the dataset. To start with, in order to deal with the few missing values that existed in random pixels at random intervals, we resorted to linearly interpolating the last time step (if it is missing) and then applied a second-order polynomial interpolation for the rest of the missing values within each pixel. This resulted in a smooth and reasonable time series where the missing values previously were. However, at this point, the InSAR dataset had two main problems:

1.　It was not equally temporally separated.
2.　It had a couple large temporal gaps.

These problems, if left unaddressed, can severely affect the performance and efficiency of a machine learning model trained on the dataset as it would be given incomplete and irregular information [90]. Therefore, not only did we have to solve these problems, but we also had to solve them in a way that maintained the inherent structure and patterns within the dataset. It is for this reason that we resorted to data augmentation. Specifically, we applied a smoothing spline using SciPy's implementation [91], which is based on the following equation:

$$\sum_j [w_j(g(x_j) - y_j)]^2 \leq s \qquad (3)$$

where $g(x)$ is the smoothing spline function, $w_j$ represents the associated weights, and $s$ is the smoothing coefficient that controls the smoothness of the resulting curve and the closeness of the approximation of the data.

While all pixels shared the same general pattern of a cyclical decreasing deformation, choosing the same smoothing coefficient for all of them was not optimal since each pixel had a slightly different behavior when it came to the actual deformation values involved. Some pixels needed more smoothing than others. Hence, we enforced an "up-down" limit which limited the amount of minima and maxima. We chose a value of 8 since the time series covered a period of approximately 4 years, which means that it was to be expected that the deformation would cycle through roughly a total of 8 local minima and maxima due to seasonal groundwater level changes [92]. Therefore, for each pixel, the smoothing coefficient was set to 100 and incremented by 50 until the up-down limit was reached. An example of the resulting smoothing spline interpolation is shown in Figure 5. As can be seen, the two large temporal gaps (start of 2018 to mid-2018 and late 2018 to mid-2019) were interpolated in a way that captured the data's natural trend. After the smoothing spline was produced for each pixel, we used a sampling period of 6 days to create an equally temporally separated time series. The sampling period was chosen to be 6 days since that aligned with the current Sentinel-1 revisit time [93]. This resulted in our dataset being augmented from 88 images to 299 images of $121 \times 201$ pixels.

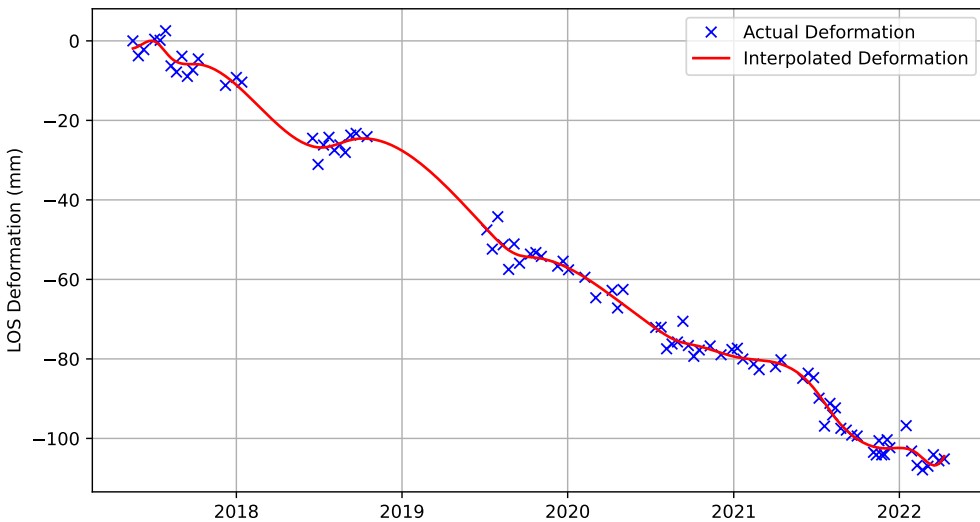

**Figure 5.** A smoothing spline interpolation for a pixel corresponding to i = 40 and j = 70.

We performed a train test split with a training percentage of 90% and 10% reserved for testing, given that our dataset was not too large. Additionally, we built a baseline model in order to be able to compare our machine learning models' results. The baseline model had to be computationally inexpensive and not too complex, so we used a linear baseline model where a linear fit based on least squares was performed for each pixel. The error metric used was the mean squared error (MSE). For each time step, the MSE was calculated for the entire image, and the average MSE over the entire range was taken to be the error.

Then, the InSAR dataset was normalized using scikit-learn's MinMaxScaler function [94], which scales the dataset to a range of 0–1 to facilitate use with machine learning models. An input $X$ and output $Y$ were created for each of the train and test sets. The input corresponded to the previous InSAR image, and the output corresponded to the next InSAR image.

The geothermal injection and production data were also preprocessed in a similar fashion. The monthly amounts were converted to daily amounts by dividing by the number of days in the given month. A train test split was applied with the same training percentage, and the inputs were created by taking the injection and production amounts of the day-of the previous InSAR image.

### 2.4. Machine Learning Models

In developing the suitable machine learning models for the deformation prediction problem, it was important to consider two crucial aspects of our dataset. First, the model had to learn the inherent spatial patterns of the InSAR dataset. This would make it more efficient at handling data pertaining to the specific region of The Geysers. Second, the model had to learn the temporal patterns for each pixel. This would make it more adept at predicting the precise deformation values of each pixel in the image. Therefore, we decided to incorporate two main layers in our machine learning models: convolutional neural networks (CNNs) and long short-term memory (LSTM) networks.

CNNs have been proven to be very powerful at processing and analyzing images [95–98]. They can even be used for time series prediction and classification [99,100]. While several CNN model architectures exist, the main architecture involves a convolutional layer, a pooling layer, and a fully connected layer [101]. These layers work together to extract the different features from the images and output a prediction or classification [102,103]. A thorough explanation of a CNN's functionality and intricacies was given by Albawi et al. [104].

LSTM networks are recurrent neural networks that specialize in retaining crucial information over long time intervals through the use of gates that govern which information to keep and which to discard. This makes LSTM networks a powerful tool that is suitable for time series prediction [105–108]. An in-depth review of LSTM networks and their applications was carried out by Yu et al. [109].

In order to build and train our machine learning models, we used Tensorflow, which is an open-source Python package for machine learning [110]. We began by building a machine learning model that only relied on InSAR data for its inputs, called model **A**. We display the model architecture in Figure 6. The model consisted of a total of 8 layers. The first layer was the input layer that took in the previous InSAR images one at a time with a shape of 121 × 201, with one channel that represented the LOS deformation. The images were then fed into a time-distributed 2D CNN layer with 16 filters and a kernel size of 3. Since we were using one image at a time (a look_back of 1), this layer was completely equivalent to a regular 2D CNN layer. The result was 16 feature maps with a slightly smaller resolution (119 × 199) which were then passed on to the 2D max pooling layer with a pool size of 2. This reduced the spatial dimensions of the feature maps to 59 × 99, thereby increasing the model's robustness and efficiency. Each feature map was then reshaped into a single vector in preparation for the LSTM layer that had 64 units. The output was then flattened in order to remove the first dimension, resulting in a vector with a dimension of 64. This was then passed on to a dense layer with 24,321 (121 × 201) units, and the output was finally reshaped into the original image size of 121 × 201.

Another model was built that essentially used InSAR data as well as geothermal injection and production data as inputs, called model **B**. This model's architecture is displayed in Figure 7. The layers dealing with the InSAR data were the same as model **A**'s. The only difference was the incorporation of the geothermal data into the model by means of two additional separate input layers. These layers took the injection and production amounts on the same date as the previous InSAR image and passed them along to a dense layer with 100 units. The resulting vectors were then concatenated with the flattened InSAR vector and passed on to the dense layer with 24,321 units followed by the reshape layer to bring the output back to the original image size of 121 × 201. Both models used a batch size of 64 and were trained for 200 epochs.

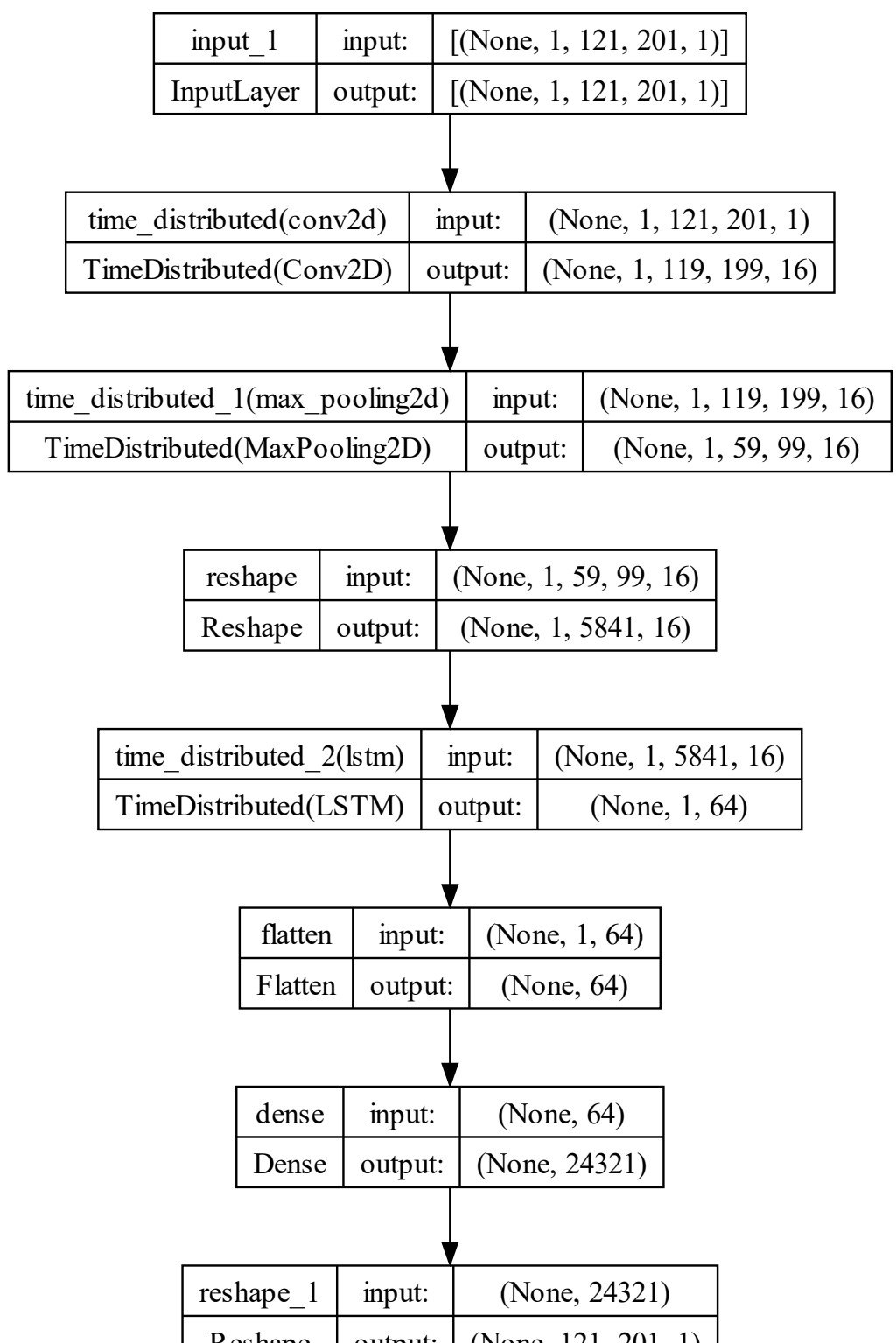

**Figure 6.** Model **A**'s architecture, including the input and output shapes.

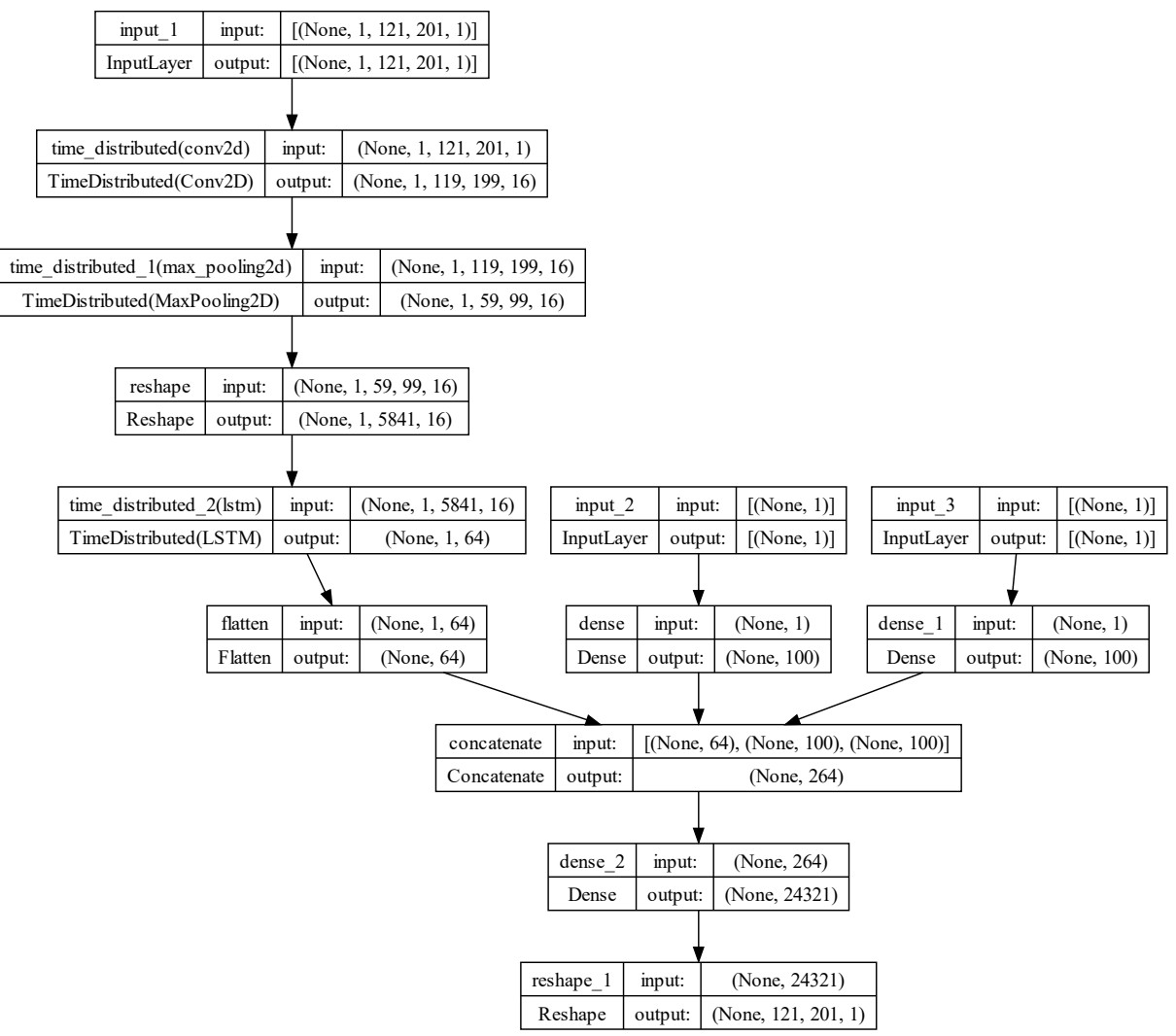

**Figure 7.** Model **B**'s architecture, including the input and output shapes.

## 3. Results

### 3.1. Correlation Results

We computed the Pearson and Spearman correlation coefficients for the seismicity count and injection amounts. The calculations were performed for both the monthly counts and the annual counts as well as for different magnitude thresholds. The results are shown in Table 1.

**Table 1.** Table showing the correlation coefficients for the monthly and annual seismicity and injection counts for different thresholds.

| Magnitude Threshold | Pearson (Monthly) | Spearman (Monthly) | Pearson (Annual) | Spearman (Annual) |
| :---: | :---: | :---: | :---: | :---: |
| 0.0+ | 0.58 | 0.64 | 0.75 | 0.78 |
| 0.5+ | 0.61 | 0.65 | 0.80 | 0.79 |
| 1.0+ | 0.65 | 0.70 | 0.87 | 0.88 |
| 1.5+ | 0.73 | 0.70 | 0.91 | 0.81 |
| 2.0+ | 0.59 | 0.55 | 0.80 | 0.67 |
| 2.5+ | 0.40 | 0.39 | 0.65 | 0.48 |
| 3.0+ | 0.16 | 0.19 | 0.30 | 0.16 |

It can be seen that the correlation coefficients increased as the magnitude threshold increased up to a threshold of 1.5, where the maximum values were achieved for most cases. Beyond this threshold, the correlation coefficients seem to steadily decrease. This indicates that the amount of injected water in the wells was most correlated with the resulting magnitude 1.5–2.0 earthquakes. For magnitude thresholds 2.5 and greater, the correlation coefficients sharply decreased. This is attributed to the fact that there are far less earthquakes falling within that range than there are in the lower magnitude ranges.

We explored the correlation further for the case of 1.5+ by plotting the monthly seismicity as a function of the monthly injection amount in Figure 8. A linear fit was added to highlight the inherent linearity between the variables. We also plotted the annual version of the same plot in Figure 9, where the linear relation is even stronger, as demonstrated by the higher Pearson correlation coefficient. What this indicates is that, to a certain degree, the induced seismicity at The Geysers is predictable, given the injection amount for that time period. While these low-magnitude earthquakes do not immediately cause severe damage, the compounding effect of them over the years definitely does. It is for this reason that we turned our attention to characterizing and predicting the resultant deformation.

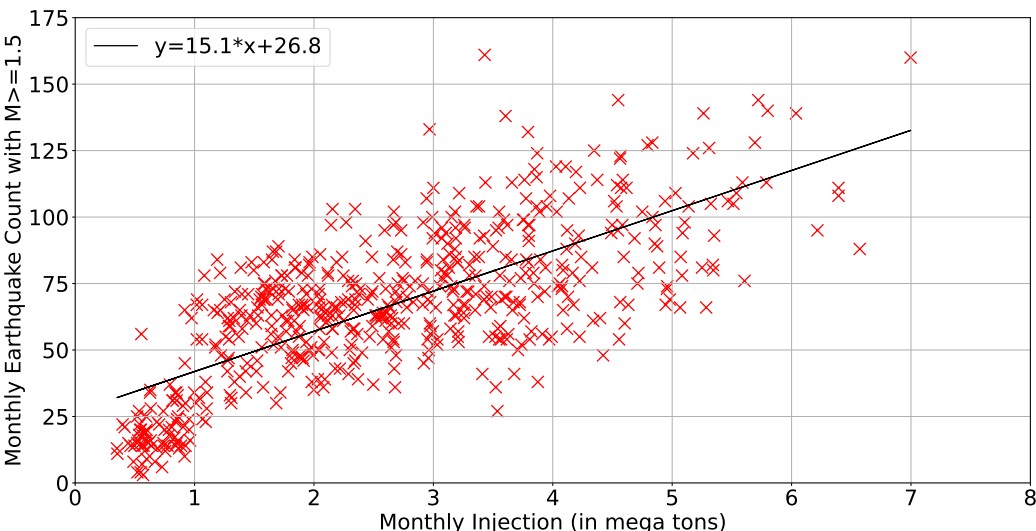

**Figure 8.** Scatter plot of the monthly seismicity and injection with a linear fit.

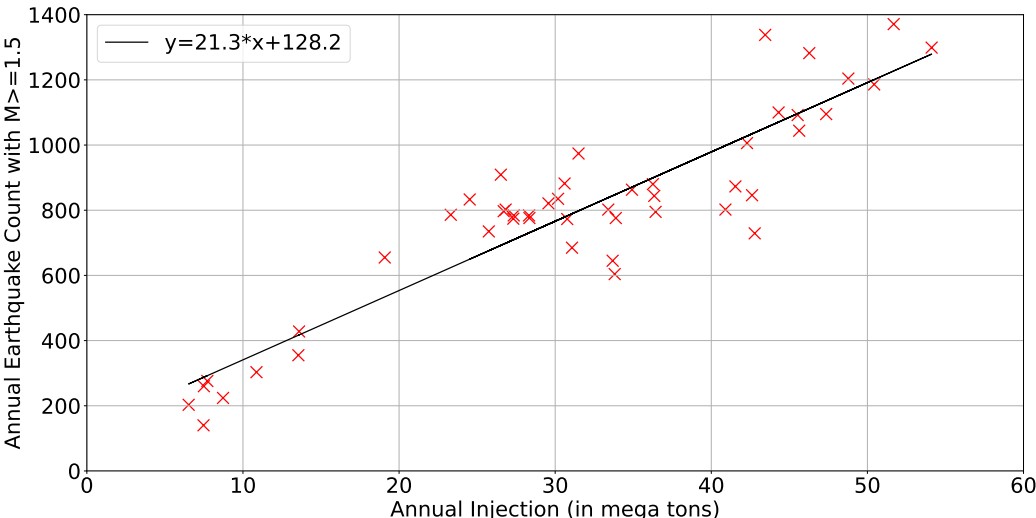

**Figure 9.** Scatter plot of the annual seismicity and injection with a linear fit.

### 3.2. Machine Learning Results

The linear baseline performed relatively well, achieving an MSE of 8.79 and 7.79 for the train and test set, respectively. It is important to note that there was no actual training involved with the linear baseline model. The test error being slightly lower than the train error is a reflection of the fact that the test set was simply more linear than the train test. This is because the test set was smaller than the train test and had less deviations from the linear model.

Due to the stochastic nature of machine learning models, it was necessary to run the models several times and take an average MSE to get a clearer idea of the actual performance of the model. Therefore, we ran each model 20 times and found that model **A** achieved a $5.95 \pm 0.27$ and $6.40 \pm 0.11$ MSE for the train and test sets, respectively, while model **B** achieved a $5.54 \pm 0.46$ and $6.03 \pm 0.12$ MSE for the train and test sets, respectively. The errors for the linear baseline model and the two machine learning models are shown in Table 2.

**Table 2.** Table showing the train and test errors of the baseline and machine learning models.

|  | Linear Baseline | Model A | Model B |
| --- | --- | --- | --- |
| Train MSE | 8.79 | $5.95 \pm 0.27$ | $5.54 \pm 0.46$ |
| Test MSE | 7.79 | $6.40 \pm 0.11$ | $6.03 \pm 0.12$ |

While both models were able to improve upon the linear baseline model, model **B** outperformed the other two, as evidenced by having the lowest average error on both the train set and the test set. The relatively low standard deviations showcase the stability and reliability of the models in reproducing the same results. Additionally, we aimed to visualize the performance of the models, and thus we plotted the first three images from the test set along with the models' predictions as well as the resulting residuals. The plots for model **A** are shown in Figure 10, and the plots for model **B** are shown in Figure 11.

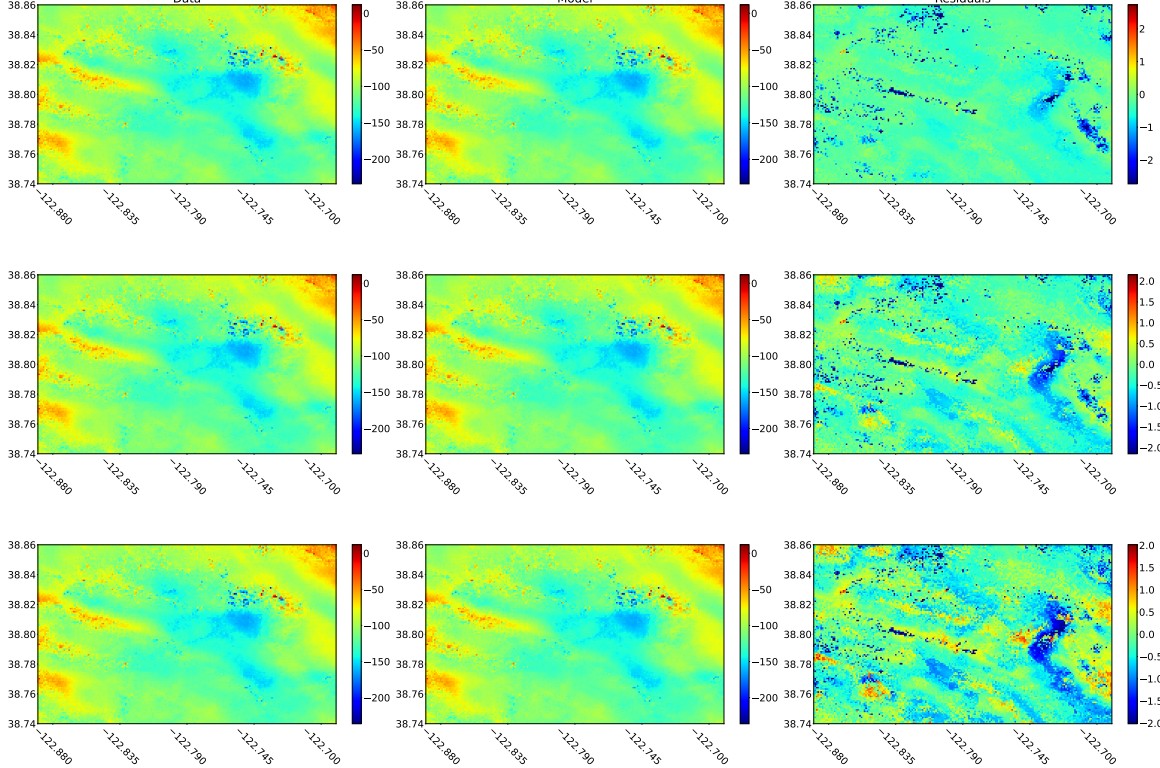

**Figure 10.** Model **A**'s prediction of the first three images from the test set along with the residuals.

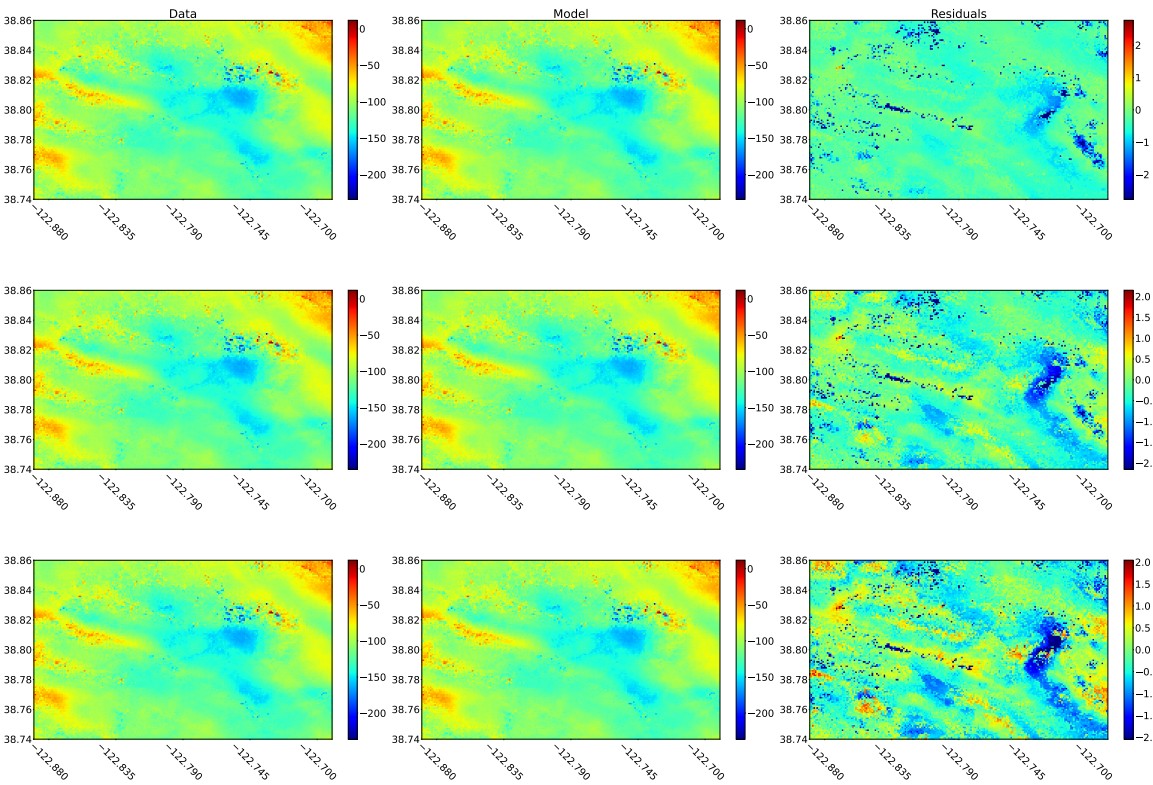

**Figure 11.** Model **B**'s prediction of the first three images from the test set along with the residuals.

Visually, both models did a good job at predicting the true data and replicating the general deformation patterns. The residuals were minor and seemed to be concentrated over the geothermal field itself. Overall, the models performed well at capturing the subsidence occurring at The Geysers and were better suited for the task than a linear baseline model, as shown by the error differences. However, it must be noted that the models were only predicting the next time step which corresponded to the deformation 6 days into the future. Attempting to predict values beyond that would cause the errors to quickly rise as each new prediction was used as input for the next prediction.

In order to further visualize the performance difference between the models, the actual and predicted deformation images from the test set were averaged and plotted on a scatter plot with a linear fit included in Figure 12. While both models generally followed the y = x line in this plot, it is model **B** that did a slightly better job at following that line, owing to its improved predictive power.

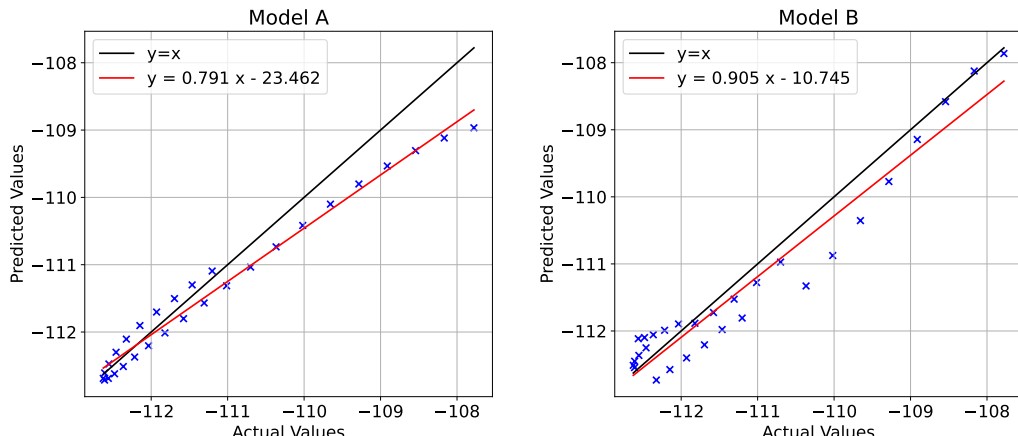

**Figure 12.** Scatter plots of actual and predicted values for the test set.

## 4. Discussion

### 4.1. Correlation and Machine Learning

The correlation between injection and seismicity at The Geysers geothermal field is well documented in the literature. To start with, Majer et al. [68] looked at this very correlation following the Santa Rosa Injection Project and found that earthquake activity had a good temporal correlation with the injection rates. Additionally, it was noted that this was especially true for seismic events with magnitudes greater than or equal to 1.5, and it was found that for earthquakes larger than magnitude 3.0, the correlation was not as present, which is in line with the findings in this study. Similar results were found by Batini et al. [111], who looked at the Larderello–Travale steam field in Italy and concluded that the injection-induced seismicity has a magnitude ceiling of around 2.0, after which correlations with injection begin to decrease. Moreover, Leptokaropoulos et al. [112] studied the correlation at two specific injection wells at The Geysers for different seismic response delays covering a time period of 7 years from 2007 to 2014. The results showed a clear positive correlation between the injection and seismicity rates and a peak correlation for a seismic delay of about 2 weeks. Our results extend the correlation analysis at The Geysers geothermal field to cover recent times and further corroborate previous researchers' works. Furthermore, this study motivates the incorporation of geothermal injection and production data into machine learning models aimed at predicting future deformations, as the data serve as a proxy for earthquake occurrence and associated subsidence.

Deformation prediction through the use of machine learning models applied to InSAR data has been performed before. To begin with, Ma et al. [113] applied a deep convolutional neural network to predict short-term deformation at the Hong Kong International Airport and found that the resulting errors were low compared with ground observations and demonstrated the effectiveness of using such a model in short-term deformation prediction. Moreover, Liu et al. [114] developed a heterogeneous LSTM network in an attempt to capture complex nonlinear temporal correlations and applied it over Cangzhou, China after dividing the area into homogeneous subregions. Their results showed that the model performed the best compared with other prediction models. In addition, in a proof-of-concept study, Hill et al. [115] applied a range of time series prediction tools on InSAR time series data and found that the LSTM models performed the best, especially when predicting signals with regular annual variations over the short term, as was the case for this study. In fact, in a previous study, we built an LSTM model and used it to predict deformation over the city of Madera in the Central Valley in California [116]. The results were compared to a baseline averaging model as well as a CNN model, and we found the LSTM model built to be the best. At The Geysers geothermal field, machine learning has been used to explore the spectral properties of seismic sources, which revealed changes in the faulting processes [60]. Additionally, machine learning has been used to predict ground motion resulting from induced seismicity, and the results were compared with an empirical ground motion model, showing that the neural network performed better [61]. To the authors' best knowledge, machine learning used to predict subsidence at The Geysers using InSAR has not been performed yet. Improvements in the machine learning model compared with our previous study [116] have been made. In fact, the combination of LSTM and CNN layers makes the models capable of capturing both temporal and spatial patterns. Additionally, the use of larger images gives the models more data to train on and increases its predictive capabilities. It is for these reasons that the models are able to improve upon the linear baseline model. Finally, the incorporation of geothermal injection and production data in model **B** made it more effective at predicting deformation than model **A**. This shows promise for models similar to model **B** to be built and applied at other geothermal fields experiencing subsidence.

Future work could be aimed at developing models that attempt prediction further into the future. This would be accomplished either by defining a bigger time step or building a model that is inherently trained for predicting several time steps into the future. This would give policy makers a better understanding of what the future of the geothermal field

looks like, aiding them in decision making. Furthermore, one could look into incorporating even more data into the model, such as reservoir pressures, temperature profiles, flow rates, geological data, and heat flow data. This would potentially make the model more accurate with its predictions by tailoring it to the specifics of The Geysers geothermal field. Finally, it is possible to further optimize the hyperparameters involved with the machine learning model, making it even better at predicting accurate deformation values. These methods could be an extensive grid search, a Bayesian-based optimization, a gradient-based optimization, or even an evolutionary algorithm.

*4.2. Limitations*

There are some limitations to take into account when looking at this study. To start with, we relied entirely and solely on the USGS catalog for the seismicity data in our study. However, several other earthquake catalogs exist that have slightly different techniques and protocols of processing data which ultimately lead to different earthquake parameters, such as the moment magnitude, location, origin time, and moment tensors. As a result, efforts have been made to assess catalog completeness as well as accuracy [117]. While the USGS catalog is deemed to be of good quality [118], no catalog is free of errors. However, for the purposes of our study, errors were as limited as possible since we only relied on the moment magnitude and counts of the earthquakes in the region for the correlation study. Therefore, the results of the study are generally reliable. Improvements could have been made by looking at different catalogs and combining them in a way to minimize errors and optimize the resulting linear model to obtain more accurate relations, but that fell beyond the scope of this paper.

The use of geodetic techniques such as InSAR is also accompanied by errors to carefully consider. Each InSAR image is subject to various errors from different sources, such as orbital, atmospheric, unwrapping, and decorrelation noise errors [119,120]. The LiCSBAS time series process typically reduces errors by eliminating unreliable InSAR images that exceed certain noise indices' thresholds. Furthermore, the incorporation of atmospheric data from GACOS is able to reduce the errors even further by essentially applying a tropospheric correction to the unwrapped InSAR images. Additionally, the perpendicular baselines $b_\perp$ of the interferograms used for the InSAR time series generation mostly ranged from $-50$ m to $+50$ m with a maximum of $+150$ m. These values are low enough that the interferograms are much less susceptible to errors arising from the digital elevation model [121,122]. A lower $b_\perp$ threshold would certainly increase the accuracy of the deformation values produced. However, its effects on the machine learning models' performance would be minimal. In fact, the deformation velocity values obtained were compared with Global Navigation Satellite System (GNSS) data. The closest GNSS station is P206, and its height is decreasing at a rate of $-1.464$ mm/year. The InSAR velocity map indicates a subsiding rate of $-1.1$ mm/year for P206's location. While the InSAR study does underestimate the velocity, its values can be considered as generally reliable.

*4.3. Recommendations*

The neural networks built in this study outperformed a baseline linear model in predicting future deformations when applied at The Geysers, proving the strength and effectiveness of using machine learning for land subsidence monitoring and forecasting. This is especially true for models that not only incorporate deformation data but also geothermal data for the case of The Geysers, as it makes the model more robust to deformation changes. Being able to predict accurate land subsidence values is crucial to establishing policies that would mitigate hazards associated with land subsidence. These policies would essentially dictate the amount of fluid extraction allowed and maintain a certain balance between the fluid injection and extraction amounts in order to achieve long-term sustainability and to prevent excessive land subsidence that would otherwise be hazardous [123]. It is for these reasons that we find it advisable to incorporate machine learning into hazard mitigation

models monitoring subsidence at The Geysers, as it would directly aid policy makers in making more informed decisions regarding the geothermal operations that are taking place.

## 5. Conclusions

In this paper, the correlation between seismicity and injection at The Geysers geothermal field was inspected, and the subsidence resulting from the induced seismicity was predicted. We started by plotting the seismicity and geothermal injection and production amounts from 1975 to 2023 and noticed that the injection and seismicity followed the same trend. Efforts to quantify the relation between the two culminated in computing the Pearson and Spearman correlation coefficients. The peak values were found to be for earthquakes with a minimum magnitude of 1.5, and the correlation values started to decrease after a minimum magnitude of 2.0, which suggests that injection is most correlated with seismic events lying in the 1.5–2.0 magnitude interval. A linear fit between the seismicity and injection was made for both the monthly and annual amounts, suggesting a predictable seismic behavior in the given injection amounts. However, the scope of this paper was more interested in the resulting deformation. It is for this reason that we used the results of the correlation study as motivation for geothermal data fusion with InSAR deformation data for prediction purposes.

An InSAR time series over The Geysers region was built using LiCSBAS, resulting in deformation data from mid-2017 to early 2022, as well as a velocity map showing the negative values in the area which highlight the subsidence taking place. Before applying our machine learning models, the InSAR and geothermal datasets were preprocessed, and a linear baseline model was built based on a mean squared error metric after splitting the datasets into train and test sets. Two machine learning models were developed which included LSTM as well as CNN layers in order to capture the temporal and spatial patterns of the data. The novelty of our approach is the inclusion of geothermal data as an additional input in model **B**, which improved upon model **A**, the model which did not include any geothermal input. This incremental improvement provided by model **B**, combined with the fact that the difference in computational times between the two models was negligible as well as the importance of having accurate deformation values, makes it favorable to use geothermal data in machine learning models that aim to predict deformation over geothermal fields. Overall, this paper shows the effectiveness of machine learning models, especially ones that incorporate additional geothermal data, in predicting the subsidence values at The Geysers and, given the importance of having accurate deformation values in the decision-making process of policy makers, shows the potential for use in hazard mitigation models.

**Author Contributions:** Conceptualization, J.Y. and J.B.R.; methodology, J.Y.; software, J.Y.; validation, J.Y. and J.B.R.; formal analysis, J.Y.; investigation, J.Y.; resources, J.B.R.; data curation, J.Y.; writing—original draft preparation, J.Y.; writing—review and editing, J.Y.; visualization, J.Y.; supervision, J.B.R.; project administration, J.B.R.; funding acquisition, J.B.R. All authors have read and agreed to the published version of the manuscript.

**Funding:** This research was funded under the United States Department of Energy grant DE-SC0017324 to the University of California, Davis.

**Data Availability Statement:** InSAR time series was formed using LiCSBAS, developed by Yu Morishita. The Geysers deformation data obtained from LiCSBAS were constrained to a grid spanning from X/Y: 1856/2476 (38.86039, $-122.88983$) as the top left corner to X/Y: 2056/2596 (38.74039, $-122.68983$) as the bottom right corner. Injection and production data of The Geysers were downloaded from the Department of Conservation of California (https://www.conservation.ca.gov/calgem/geothermal/manual) (accessed on 15 February 2023). GNSS data for P206 were obtained from the Jet Propulsion Laboratory (JPL) website (https://sideshow.jpl.nasa.gov/post/series.html) (accessed on 17 October 2023). InSAR velocity for P206 was obtained from point (2170,2569) in the LiCSBAS framework.

**Acknowledgments:** LiCSAR contains modified Copernicus Sentinel data (2014–2021) analyzed by the Centre for the Observation and Modelling of Earthquakes, Volcanoes and Tectonics (COMET). LiCSAR uses JASMIN, the UK's collaborative data analysis environment (http://jasmin.ac.uk) (accessed on 15 March 2023).

**Conflicts of Interest:** The authors declare no conflict of interest. The funders had no role in the design of the study; in the collection, analyses, or interpretation of data; in the writing of the manuscript; or in the decision to publish the results.

## Abbreviations

The following abbreviations are used in this manuscript:

| | |
|---|---|
| InSAR | Interferometric synthetic aperture radar |
| CNN | Convolutional neural network |
| LSTM | Long short-term memory |
| LOS | Line of sight |
| MSE | Mean squared error |
| GACOS | Generic Atmospheric Correction Online Service for InSAR |
| EGS | Enhanced Geothermal System |
| USGS | United States Geological Survey |
| ML | Machine learning |
| GNSS | Global Navigation Satellite System |

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
