# Peer review of "A Fusion of Geothermal and InSAR Data with Machine Learning for Enhanced Deformation Forecasting at the Geysers"

_land, doi:10.3390/land12111977_

Round 1

Reviewer 1 Report

Dear authors,

This manuscript presents a fusion AI framework that leverages geothermal and insar data for the improved deformation forecasting at the geysers. The manuscript first discuss and validate the correlation between the injection of water and the mild earthquake activities. Then, retrieve the ground deformation with LiCSBAS and perform the prediction by two ML architecture. Although the manuscript's structure is well-organized, and the authors have conducted thorough research, the mainly my concern is the innovation of this manuscript. The performance and effectiveness of the proposed framework are not well presentated and validated. Given that many research have discussed the correlation between the injection activities and. the earthquake. In addition, many of the models and methods (CNN and LSTM) employed are widely used across various applications, including in the geysers as you stated in the manuscript. In my opinion, doing the same research with the latest dataset is diffcult to be taken as the innovative aspect. So could you please make your statement more concise and carefully introduce your contibution or innovation? To address this concern, we suggest conducting additional experiments to highlight the innovation and advancement of your methodology and work.

Major Comments:

1. In the introduction section, please revise it to effectively introduce the research background, current situation, research gaps, and your innovations. The current version has an extensive background but lacks a clear solution provided by your research.

2. Could you please revise your InSAR experiments and display the deformation results with some other color palette like Jet? In my opinion, your deformation results may be not reliable.

3. Why did you provide the review statement in your discussion part? In addition, according to the limitation part, it seems you are questioning your work. Could you please extend your experiment according to the plan or suggestion in your discussion part? Revise the whole discussion part with some sections like validation, comparison with existing methods/works, introduction of some new findings, etc.

4. The conclusion part needs to be re-planned. Please state your true contribution clearly and don’t oversell your work.

Minor Comments:

1. It would be better to display the fault (you mentioned in the content) in the Figure 1

2. Could you please introduce or use the experiment to highlight the interpret the data in an equal interval that would be better than others during on page 9 section 2.3?

3. On page 12, can you please provide more information on why the correlation with the magnitudes larger than 2.0 is not high? Do you think it would be possible that because the number of such earthquakes is less leading to a lower correlation?

4. What does the baseline mean in Table 2?

Language is fine

Author Response

Dear reviewer,

We would like to thank you for your review and suggestions. In regards to the major comments, we added to our introduction the gaps in the machine learning applied to the study area and highlighted the novelty of the work. We changed the color palette of our InSAR images to Jet to better highlight the negative and positive values. Also, we created a scatter plot to better visualize the difference in predictive powers of the models. We added to our discussion a more focused comparison to existing works in regards to the LSTM models applied to InSAR. Finally, we highlighted the novelty of our approach in the conclusion. With regards to the minor comments, we added the major faults in the map of the area so it's clearer to the reader. We also indicated the reason for the lower correlation as you pointed out, and we made sure that the 'Linear Baseline' was made clear in Table 2.

We appreciate your remarks and suggestions that helped improve the paper.

Thank you for your review!

Reviewer 2 Report

This publication investigates the implementation of a Machine Learning model that incorporates both geothermal data and InSAR observations to predict surface deformation as a result of seismic activity. The manuscript is sound from a scientific standpoint and merits publication in Land Journal. Overall, the paper is well-written and organized, however, I have made some minor suggestions for improvement. With these revisions complete, the manuscript can be submitted to Land Journal for publication.

Comments:

- The authors noted two limitations of InSAR inherent in the SAR data used in this study: a lack of a temporal baseline and a time lag between scenes. In addition to these factors, which may have a significant impact on the built ML model, the authors do not discuss the perpendicular baseline threshold used in this study and its potential impact on the model's estimates of surface deformation. Therefore, it will be helpful if the authors include a discussion of the threshold that can be allowed to avoid any bias in the ML model's prediction of deformation.  

- Model B did not significantly outperform model A (particularly in terms of visual inspection), indicating the limited influence of geothermal data in reproducing surface deformation. The reviewer suggests the writers provide further explanation in that regard.

Author Response

Dear reviewer,

We would like to thank you for your review and suggestions. We have added the perpendicular baseline thresholds used in our study and expanded on it in the 'Limitations' section. The baseline values of our interferograms are low enough so that digital elevation model errors are minimal based on previous studies. In addition, we have made an additional visual figure (scatter plot) to highlight the difference in the models' predictive powers where it can be seen that model B's linear fit more closely matches the y=x line on a predicted vs actual plot. We have also added an explanation in the conclusion that this improvement makes it favorable to include geothermal data especially since the computational time is basically the same as well as the importance of having accurate deformation values.

Thank you so much for your review!

Reviewer 3 Report

Dear Authors,

A very interesting article on the determination and prediction of land surface deformations in a geothermal field. The field conditions made it possible to fully use the satellite interferometric synthetic aperture radar (InSAR) to model land surface deformations as a result of the use of geothermal energy by injecting and pumping water. I like the insightful approach to solving the problem and the detailed conclusions, limitations and recommendations formulated in the article. However, the issue of geodetic reference measurements should also be included in the analysis. Despite the displacement velocity (LOS) values, which ranged from 5 cm/year to 90 cm/year, convenient for determining using InSAR methods, in my opinion it was necessary to compare them with the results of classic geodetic measurements that were previously conducted by the National Geodetic Survey. My experience shows that methods such as SBAS or PSInSAR often significantly underestimate the LOS displacement and need to be corrected based on the reductions recorded by geodetic measurements. Another detailed note, the term satellite geodesy, is referred to in the article as InSAR methods. This term is associated rather with GNSS (Global Navigation Satellite Systems) methods, while InSAR (as written on page 5/162) is a remote sensing technique and should be understood as such.

Possible directions for further research. Current research results indicate the possibility of decomposing the LOS vector and determining the horizontal displacement field. On this basis, it is possible to determine the elements of a flat deformation tensor, the distribution of which would indicate the zones of the greatest deformations of the land surface and possible damage to objects and infrastructure.

I recommend the article for publication after the above corrections.

Author Response

Dear reviewer,

We would like to thank you for your suggestions and notes. We incorporated geodetic measurements as per your suggestion. The closest GNSS to the Geysers is P206 so we looked at the height change and found it to be -1.46mm/year. Our InSAR velocity map showed a velocity of -1.1mm/year which does underestimate the true velocity, but we deemed it to be close enough to the real value as it wouldn't drastically affect the machine learning model's performance. Additionally, we fixed our reference to InSAR methods as 'remote sensing techniques' rather than 'satellite geodesy' as you correctly pointed out.

Thank you so much for your review!

Round 2

Reviewer 1 Report

Dear Authors,

Thanks for your effort to address my comments. Although I am still thinking about the reliability of your InSAR result as there seems to be an effect from the serious APS and topo-related APS, we will give you a pass because you only focus on the geothermal area. I have no more comments.